# Metastatic Uterine Adenocarcinoma in a Sable Antelope (*Hippotragus niger*)

**DOI:** 10.3390/vetsci9070339

**Published:** 2022-07-04

**Authors:** Louise van der Weyden, Anien Bezuidenhout, Erna van Wilpe, Nicolize O’Dell

**Affiliations:** 1Wellcome Sanger Institute, Wellcome Genome Campus, Hinxton, Cambridge CB10 1SA, UK; lvdw@sanger.ac.uk; 2Department of Paraclinical Sciences, Faculty of Veterinary Science, University of Pretoria, Onderstepoort, Pretoria 0110, South Africa; anien.bez6@gmail.com; 3Department of Anatomy and Physiology, Faculty of Veterinary Science, University of Pretoria, Onderstepoort, Pretoria 0110, South Africa; erna.vanwilpe@up.ac.za; 4Laboratory for Microscopy & Microanalysis, Department of Physics, Faculty of Natural & Agricultural Sciences, University of Pretoria, Hatfield, Pretoria 0028, South Africa; 5Centre for Veterinary Wildlife Studies, Faculty of Veterinary Sciences, University of Pretoria, Onderstepoort, Pretoria 0110, South Africa

**Keywords:** antelope, cytokeratin, electron microscopy, uterine adenocarcinoma, metastasis

## Abstract

**Simple Summary:**

A female sable antelope with a history of gradual loss of body condition was found dead by the owner. Macroscopic examination revealed an enlarged spleen and liver that were covered in white-to-cream-coloured nodules. The uterus also showed a few small, white-to-cream-coloured nodules, with similar nodules present in other parts of the body. Microscopic analysis of sections of the uterus revealed tumour cells with an appearance that was similar those seen in sections of the other tissues. Critically, tumour cells were seen in the lymphatics within the lungs. The tumour cells in the uterus showed positive staining for cytokeratin as did the tumour cells in the sections of other tissues, confirming they were of epithelial origin. In addition, electron microscopy of the uterus and liver showed tumour cells arranged in groups with junctions present between the cells. This confirmed that the tumour cells seen in the liver were the same as those seen in the uterus and were of epithelial origin. Thus, a diagnosis was made of uterine adenocarcinoma with widespread metastasis. This is the first report of uterine adenocarcinoma in a sable antelope.

**Abstract:**

A nine-year-old intact female sable antelope (*Hippotragus niger*) with a six-week history of gradual loss of body condition was found dead by the owner and presented for autopsy. Macroscopic examination revealed an enlarged spleen and liver with the hepatic and splenic parenchyma showing extensive infiltration with firm, white to cream-coloured nodules. The uterus showed a few small, firm, well-demarcated, white-to-cream-coloured nodules in the uterine body. Similar nodules were present in the mediastinum, parietal pleura, heart, and marrow cavity of the femur. Histological analysis of the uterus revealed densely cellular neoplastic proliferations, forming nests, tubules, and acini within an abundant fibrovascular stroma. The samples from the other tissues revealed neoplastic cells with a similar appearance to those seen in the uterus, also forming nests and acini in a fibrovascular stroma. Importantly, multiple neoplastic cells were also seen in the peribronchiolar lymphatic vessels. The neoplastic cells in the uterine sections showed positive immunohistochemical labelling for cytokeratin, as did the neoplastic cells in the sections of liver and parietal pleura, confirming they were of epithelial origin. In addition, transmission electron microscopy of the uterus and liver showed neoplastic cells arranged in groups surrounded by basement membranes and interspersed with collagen fibres. Junctions were present between the cells, and junctional complexes could be seen at some cell surfaces. This confirmed that the neoplastic cells seen in the liver sample were the same as those seen in the uterine sample and were of epithelial origin. Thus, a diagnosis was made of uterine adenocarcinoma with widespread metastasis. This is the first report of uterine adenocarcinoma in a sable antelope.

## 1. Introduction

Uterine cancer is uncommon in domestic animals [1,2] in contrast to its prevalence in humans where it is the most frequent gynecological malignancy [3]. The most commonly reported uterine cancer in domestic animals is leiomyoma in a variety of species [2]. In contrast, carcinoma of the uterus is rare and apart from individual case reports in different species, such as the cat [4], dog [5], horse [6], sika deer (*Cervus nippon*) [7], and elk (*Cervus elaphus nelsoni*) [8], it has typically only been reported in rabbits and older cows at slaughter. For example, in a review of rabbit neoplastic and non-neoplastic masses submitted for histological examination, 9% of the biopsies were from tumours of the female reproductive tract and were predominantly uterine (endometrial) adenocarcinoma [9]. In addition, uterine adenocarcinoma was the most commonly reported neoplasm found in histological assessments of uterine lesions in pet rabbits in Japan following ovariohysterectomy (1035/1928 cases, 53.7%) [10]. Furthermore, uterine adenocarcinoma was classified as one of the more commonly encountered tumour types in a study of bovine neoplasms in Canadian slaughterhouses (53/1370 cases, 3.8%) [11]. In addition, there have been several case reports of uterine adenocarcinomas in pigs [12,13,14], and one study found a high prevalence of neoplastic uterine lesions in miniature pet pigs with 2/20 (10%) of the neoplastic lesions being adenocarcinomas [15].

There are only two reports of uterine tumours in antelopes to date, including uterine leiomyomas in three captive eastern bongos (*Tragelaphus eurycerus isaaci*) [16] and a uterine adenocarcinoma in a captive scimitar-horned oryx (*Oryx dammah*) [17]. The latter case presented as dystocia, and the presence of a uterine adenocarcinoma with pulmonary metastases was only found during a post-mortem examination when the oryx died 36 h after surgical removal of the dead premature calf [17]. The present report describes uterine adenocarcinoma with widespread metastasis in a sable antelope.

## 2. Case Presentation

In this report, a nine-year-old intact female sable antelope (*Hippotragus niger*) that was part of a herd with wireworm (*Haemonchus contortus*) infestation and had a six-week history of gradual loss of body condition was found dead by the owner. The sable antelope originated from a private game farm where the animals are kept in an extensive system consisting of large camps (20 ha) with natural vegetation and receiving some supplementary feeding during the dry winter months. She was born on the farm and had given birth uneventfully to six calves on a yearly basis since reaching reproductive maturity in 2008 until her death in 2014.

The relatively fresh carcass of the sable antelope was presented for post-mortem examination. External examination found no abnormal lesions present on the skin; however, the body condition was poor (scored as 2/5). Macroscopic examination of the abdomen revealed an enlarged spleen and liver with multifocal-to-coalescing, firm, white-to-cream coloured-nodules (Figure 1a). The cut surface through the hepatic (Figure 1b) and splenic (Figure 1c) parenchyma showed extensive infiltration by the firm, white-to-cream-coloured nodules with areas of necrosis also noted. On examination of the thoracic cavity, the cranioventral mediastinum was found to have a few white, small, multifocal-to-coalescing proliferative nodules. The parietal pleural surface of the mid-thoracic cavity on the left side had a small well-demarcated red-discoloured nodule with white proliferative tissue around the nodule (Figure 1d). The uterus showed no evidence of neoplastic nodules on the serosal surface; however, upon dissection there were a few small, firm, well-demarcated, white-to-cream-coloured nodules present in the uterine body that almost resembled involuted caruncles macroscopically (Figure 1e). Examination of the long bones revealed serous fat atrophy in the marrow cavity of the femur with a relatively large white nodule present in the proximal part in the red marrow (Figure 1f). A peripheral blood sample from the ear was taken to generate a blood smear, and the results were within normal limits. A stool sample was taken from the rectum for a faecal flotation test and was positive for *Strongyloids*-type ova (scoring 2+).

Tissue samples from the uterus, heart, lung, pleura, mediastinum, spleen, liver, lymph node, and bone marrow were fixed in 10% buffered formalin, embedded in paraffin wax, and sectioned for histopathological evaluation. Sections (4 μm) were stained with haematoxylin and eosin (HE). Immunohistochemistry (IHC) was performed on selected sections to evaluate the expression of cytokeratin using a monoclonal mouse anti-human pan-cytokeratin antibody (monoclonal cytokeratin AE1/AE3; Dako, code M3515). After dewaxing, sections were treated with 3% hydrogen peroxide in methanol for 15 min, then enzymatic epitope retrieval using a protease solution (Sigma-Aldrich, code no. P5147-SG) followed by incubation with the primary cytokeratin antibody at 1:400 dilution for one hour. Thereafter, the BioGenex Super Sensitive^TM^ Polymer-HRP IHC Detection System (BioGenex, code no. QD420-YIKE) was used according to the manufacturer’s instructions. Sections were further incubated with a DAB chromogen (Dako, code K3468) for 1–2 min, followed by routine counterstaining in Mayer’s haematoxylin, washing in running tap water, dehydration in alcohol and xylene, and coverslipping. All sections were examined by light microscopy by a veterinary pathologist. Standard transmission electron microscopy (TEM) was performed on selected tissue samples (uterus and liver). Briefly, the formalin-fixed samples were post-fixed in 1% osmium tetroxide in Millonig’s buffer, dehydrated through a graded ethanol series, infiltrated with a propylene oxide/epoxy resin mixture and embedded in absolute resin. Ultra-thin resin sections were stained with uranyl acetate and lead citrate and examined in a Philips CM 10 transmission electron microscope operated at 80 kV.

Histological analysis of all the affected organs were performed, and all the samples revealed similar changes that were characterised by densely cellular proliferations of neoplastic cells forming sheets, nests, and occasionally tubules and acini. The epithelial appearance in some areas came as a surprise since at necropsy, the most affected organ was the spleen, and at the time, the neoplasm was suspected to be lymphoid in origin. The uterine lesions that were originally interpreted as involuted caruncles revealed similar neoplastic cells that appeared to originate from the uterine glandular epithelium and displayed the same pattern as the other neoplastic foci (Figure 2a). In the liver, and similarly in the spleen, the parenchyma showed multiple infiltrates of neoplastic cells forming nests, tubules, and acini within a fibrovascular stroma, compressing the hepatocytes to one side (Figure 2b) and obliterating the splenic red pulp (Figure 2c). In the lungs, similar appearing neoplastic cells were multifocally present, resulting in areas of consolidation (Figure 2d). Importantly, multiple neoplastic cells were also noted in the peribronchiolar lymphatic vessels (Figure 2e). The nodule from the thoracic pleura (Figure 2f) as well as the epicardium and myocardium revealed infiltration by multiple neoplastic cells (Figure 2g). In the femur, the bone marrow showed multiple neoplastic cells and a comparative lack of adipocytes (Figure 2h). At higher magnification the neoplastic cells were medium sized with scant clear cytoplasm and indistinct cell borders. The nuclei were round to oval, clumped to vesicular, and hyperchromatic with one or more indistinct basophilic nucleoli (Figure 2i).

The mitotic rate was extremely high ranging from 15 mitoses per high power field in the primary tumour to almost 50 mitoses per high power field in some of the metastatic foci. Small areas of necrosis were occasionally interspersed between the neoplastic cells. Given the histopathological similarity of the neoplastic foci present in the different tissues to those seen in the uterus, it was suspected that the primary tumour was a uterine adenocarcinoma with widespread metastasis.

Due to the scant cytoplasm and a mild degree of autolysis resulting in occasional cell distortion in some areas of the tissues, IHC and TEM were elected to confirm the epithelial nature of the neoplastic cells. In the uterine sections, the majority of the neoplastic cells showed strong cytokeratin-specific positive cytoplasmic labelling (Figure 3a) as did the neoplastic cells in the liver and pleural surface (Figure 3b,c), confirming they were all of epithelial origin.

TEM of the uterus and liver showed neoplastic cells with a similar appearance (as the neoplastic cells from all the metastatic tissue sites showed a similar appearance, the liver was chosen as a ‘representative’ tissue for analysis). The uterus showed neoplastic cells arranged in groups surrounded by basement membranes and interspersed with collagen fibres, adjacent to normal appearing endometrial glands. Intermediate filaments and lipid droplets were sometimes seen in the cytoplasm, attachments were present between the cells, and junctional complexes could be discerned at some cell surfaces (Figure 4a–c). The neoplastic cells in the liver sample were arranged in groups that were interspersed with collagen fibres and fragmented basement membranes surrounded the cell groups. Similar to the neoplastic cells in the uterus, intermediate filaments and lipid droplets were sometimes seen in the cytoplasm, and attachments were present between the cells (Figure 4d–f). This confirmed that the neoplastic cells seen in the liver sample were of epithelial origin and were the same as those seen in the uterine sample. Thus, a diagnosis was made of uterine adenocarcinoma with widespread metastasis.

## 3. Discussion

The sable antelope (*Hippotragus niger*) is a gramivorous and folivorous savanna woodland species that is found in East and Southern Africa [18]. With threats, such as unregulated hunting, civil war, habitat loss/degradation, expansion of pastoralism, and disease, many antelope species have been placed on the International Union for Conservation of Nature’s (IUCN) “red list”, including the sable antelope [19]. Although currently designated by the IUCN as “least concern”, the free-ranging/wild sable antelope population faces on-going threats due to poaching and significant loss of habitat [19]. Thus, knowledge we can gain in understanding the health concerns that face this species is important to aid the survival of the limited populations that exist.

There have only been two reports of tumours in sable antelope in the literature to-date, specifically an oropharyngeal teratoma [20] and a BPV-1-associated cutaneous fibropapilloma [21]. In the present study, we report a sable antelope with metastatic uterine adenocarcinoma. As the uterus only showed a few nodules, suspected to be involuted caruncles, relative to the massive tumour burden seen in other tissues such as the spleen, it was not thought to be the origin of the tumour at the time of macroscopic examination. However, histopathological analysis together with IHC and TEM confirmed the epithelial origin of the neoplastic cells. The neoplastic cells were positive for cytokeratin, similar to that reported for pan-cytokeratin immunohistochemical staining of uterine adenocarcinoma in rabbits and cats [22,23,24], as well as a case report of a uterine adenocarcinoma in a miniature pig [14]. Immunohistochemical studies of cytokeratin expression (pan CK, CK1, CK5, CK6, CK7, CK8/18, CK10, CK13, and CK14) in bovine uterine epithelium found a consistent pattern between normal endometrium and primary well-differentiated uterine adenocarcinomas and their metastases [25]. This is consistent with that found in humans where epithelium-derived tumours have been reported to maintain a cytokeratin expression typical of their non-transformed counterparts [26]. However, immunohistochemical analysis of specific cytokeratins in feline uterine adenocarcinoma have shown mixed results for alterations in CK7 and CK20 positivity relative to normal endometrial epithelium (which is CK7+/CK20+) [24,27,28].

In this case study, several small nodules were present in the body of the uterus, and metastasis was observed in all tissues examined, specifically the spleen, liver, lung, heart, mediastinum, pleural cavity, and bone marrow. In cattle, uterine adenocarcinomas tend to be solitary and develop in the uterine horn, manifesting as discrete, firm enlargements of the uterus [2]. Invasion into the veins and/or lymphatics is frequently present with metastasis to other parenchymal organs, such as the lung and liver, also observed [2]. In a study of bovine neoplasms encountered in Canadian slaughterhouses, metastatic lesions of uterine adenocarcinoma accounted for the majority of neoplasms found in the lung [9], and there have been case reports of uterine adenocarcinoma in cows presenting with metastasis to a range of sites including the lung, liver, mesentery, and ovaries [28,29,30,31]. There are also three case reports of pigs [12,13,14] and two case reports of deer [7,8] with uterine adenocarcinoma that have metastasized to the lymph nodes, liver, lungs, and/or ovaries. Similarly, two case reports of uterine adenocarcinoma in domestic cats reported the presence of metastases to the abdomen [4,32]. There is also a case report of a pet rabbit with a pathologic fracture of the femur due to uterine adenocarcinoma metastasis [19].

In dogs and cats, the recommended treatment for uterine adenocarcinoma is surgical removal, typically via an ovariohysterectomy. This treatment has had both successful outcomes (clinically healthy and no signs of metastasis when last checked at ≥21 months post-surgery) [33,34] and unsuccessful outcomes (recurrence, followed by metastasis and death) [35,36]. Ovariohysterectomy has also been successfully used to treat uterine adenocarcinoma in pigs (miniature pigs, pot-bellied pigs, and production-sized pigs) [37,38]. Interestingly, while dogs, cats, and rabbits may be protected from uterine neoplasia by the frequent practice of ovariohysterectomy at a young age, there have been reports of adenocarcinoma developing from the uterine stump [24,39,40]. In addition, adjunct chemotherapy has also successfully been used in some cases, such as epirubicin treatment in a dog [41]. Treatment of uterine adenocarcinoma in other species has not been reported, presumably due to the tumour typically only being detected at necropsy (either due to death of the animal from the uterine adenocarcinoma with or without metastasis, such as in this case report, or an incidental finding after death of the animal due to other circumstances, such as cows at an abattoir).

Studies on uterine adenocarcinoma in humans have shown it is a multifactorial disease, with many underlying factors playing a role in its development, including exposure to excess oestrogen and/or a relative lack of progesterone (as oestrogen, in contrast to progesterone, stimulates the rapid growth of endometrial cells), and certain dietary factors, such as consumption of foods high in animal fats and sugars (as opposed to diets high in vegetables and fruits) [42]. In humans, the known risk factors for uterine adenocarcinoma include obesity (a high body mass index and waist-to-hip ratio) being associated with an increased risk of incidence, and parity being associated with a reduced risk (relative to nulliparous women) [43]. However, the pathogenesis and/or the risk factors leading to development of uterine adenocarcinoma in animals is unknown due to the low number of cases reported and/or not all organs being examined histopathologically at the time of death (leading to an underestimation of the true incidence). However, in both humans and animals, uterine adenocarcinoma is generally seen in older females [2], as was the case with the sable antelope in this report (sable antelopes have a lifespan of up to 16 years in the wild). Nevertheless, there have been reports of this tumour type in young animals, such as a 10-month-old golden retriever [41], four domestic shorthaired cats <1 year of age [32], and two Persian cats at 4–5 years of age [34], suggesting that young age should not be an excluding factor for a differential diagnosis of uterine adenocarcinoma.

## 4. Conclusions

This is the first report of a uterine adenocarcinoma in a sable antelope and adds to the knowledge of the range of tumour types that can be developed by this species. In addition, the present case is an additional contribution to the very few histopathological reports of this tumour type in domestic and wild animals. Given the invasive nature of uterine adenocarcinomas and as such patients frequently have spread to the lungs and other organs at the time of presentation, metastatic uterine carcinoma should be added to the differential diagnosis of sable antelopes showing a generalised poor condition. It is hoped that further investigations of sable antelopes in wildlife reserves or zoological facilities will add to a better knowledge and understanding of the tumour types and pathologies developed by this species to aid in diagnosis and therapeutic options.

## Figures and Tables

**Figure 1 vetsci-09-00339-f001:**
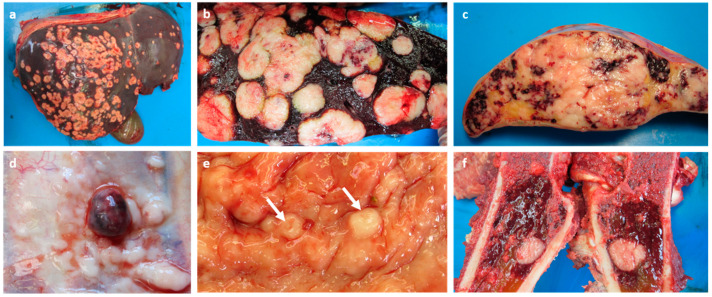
Macrosopic photos of the neoplastic masses present in the organs of the sable antelope at necropsy. (**a**) Liver with multifocal-to-coalescing, firm, white-to-cream-coloured hepatic metastatic nodules with some central areas of necrosis present. The right and quadrate lobes appeared to be the worst affected parts. Cut surface of the liver (**b**) and spleen (**c**) showing multifocal-to-coalescing, firm, white-to-cream-coloured splenic metastatic nodules (with some areas of necrosis) invading the parenchyma. (**d**) A small well-demarcated red nodule surrounded by white proliferative tissue on the parietal pleural surface of the left-side of the mid-thoracic cavity. (**e**) A few firm, well-demarcated, white-to-cream-coloured nodules (arrows) in the uterine body. (**f**) Cut surface of the femur with one solitary white-to-cream-coloured nodule present in the proximal end of the red marrow.

**Figure 2 vetsci-09-00339-f002:**
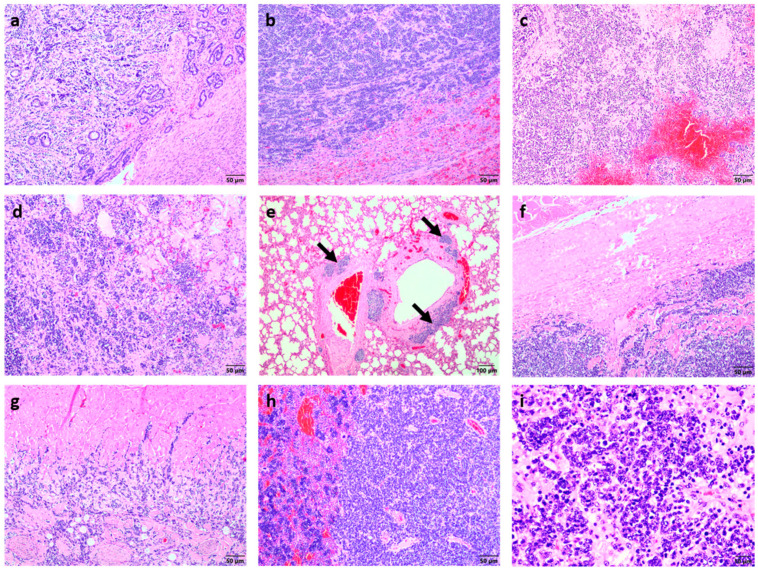
Histopathological appearance of the tumour cells in the different tissues. (**a**) The neoplastic cells in the uterus formed dense cellular nodules of various sizes consisting of nests, tubules, and acini of neoplastic cells within an abundant fibrovascular stroma (H&E stain, bar = 50 μm). (**b**) The hepatic parenchyma contained multiple neoplastic infiltrates forming nests, tubules, and acini within a fibrovascular stroma, compressing the hepatocytes to the side (H&E stain, bar = 50 μm). (**c**) The splenic red pulp was completely obliterated by neoplastic cell infiltration (H&E stain, bar = 50 μm). (**d**) The pulmonary parenchyma showed multifocal clusters of neoplastic cells similar to those seen in the liver and uterus, resulting in areas of consolidation (H&E stain, bar = 50 μm) with (**e**) multiple neoplastic cells in the peribronchiolar lymphatic vessels (arrows) (H&E stain, bar = 100 μm). (**f**) The nodule from the thoracic parietal pleura showed proliferation of multiple neoplastic cells (H&E stain, bar = 50 μm). (**g**) The epicardium and myocardium were similarly infiltrated by the neoplastic cells (H&E stain, bar = 50 μm). (**h**) The bone marrow had a lack of adipocytes with multiple neoplastic cells present (right-hand side; H&E stain, bar = 50 μm). (**i**) Higher magnification of the neoplastic cells revealed medium-sized neoplastic cells with scant cytoplasm, hyperchromatic nuclei, and a very high mitotic rate (H&E stain, bar = 10 μm).

**Figure 3 vetsci-09-00339-f003:**
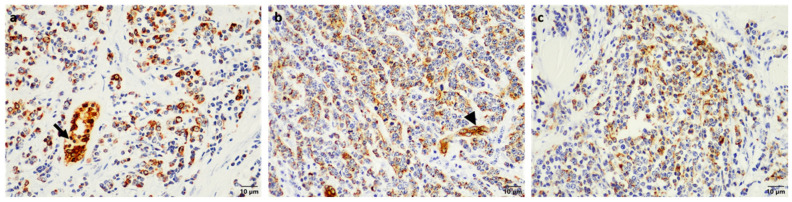
Immunohistochemical staining for cytokeratin in the neoplastic cells. Neoplastic cells in the (**a**) uterus, (**b**) liver, and (**c**) pleura showing strong cytoplasmic cytokeratin-positive labelling by immunohistochemistry (bar = 10 μm) with the normal uterine glands (arrow) and hepatic bile ducts (arrowhead) serving as internal controls.

**Figure 4 vetsci-09-00339-f004:**
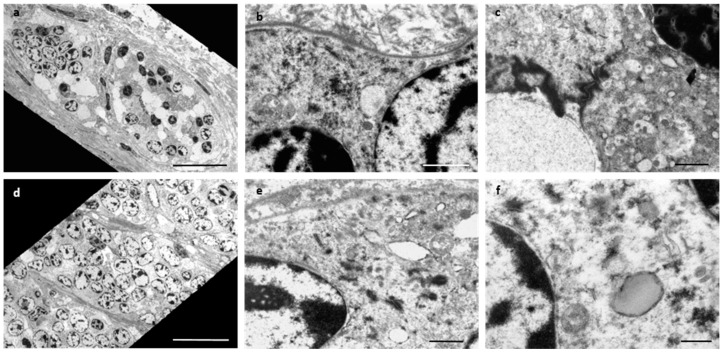
Transmission electron microscopy of the neoplastic cells in the uterus and liver. The neoplastic cells in the uterus (**a**–**c**) and the liver (**d**–**f**) were arranged in groups (**a**,**d**) surrounded by basement membranes and interspersed with collagen fibrils (**b**). Intermediate filaments (**e**) and sparse lipid droplets (**f**) were evident in the cytoplasm. Attachments (**b**,**e**,**f**) were present between cells and junctional complexes (**c**) and appeared at some cell surfaces. Bar = 20 µm (**a**,**d**); 1 µm (**b**,**c**,**e**); 500 nm (**f**).

## Data Availability

Not applicable.

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
