# Peer review of "Metastatic Uterine Adenocarcinoma in a Sable Antelope (Hippotragus niger)"

_vetsci, 2022, doi:10.3390/vetsci9070339_

Round 1

Reviewer 1 Report

This case report is a very detailed examination of metastatic uterine adenocarcinoma in a sable antelope. The autopsy examination is very comprehensive with multiple tumour sites investigated. Additional efforts (IHC and TEM) were appropriately used to examine fine detail of the tumours.

Overall, a very thorough report with sound conclusions as to tissue of origin.

Author Response

We thank the reviewer for their appreciation of our manuscript and are delighted they believe it represents a “very thorough report with sound conclusions”.

Reviewer 2 Report

The manuscript is well written.  It is true that there are very few published reports of uterine neoplasia in ungulates in general, and specifically in antelopes, therefore this case is valuable, but this is only one case.  The authors have done a great job of describing the findings and using CK IHC and TEM to confirm origin of metastatic lesions.  However, it reads as a very long detailed histopathology report, and other than a confirmed metastatic uterine carcinoma in one sable antelope, there is little additional information in the manuscript.

The authors provide few details about the history of the female: is this a zoo and thus intensely managed, potentially contracepted? or a reserve where she presumably bred every year? Because the reproductive life history can influence risk factors please add some the missing history about this female

Line 256- in order for the reader to be able to appreciate that 8yr is an older female, please add the median life expectancy for this species

In the discussion the authors compare histological findings in tissues in the tissue as well as metastases.   Other IHC such as hormone receptors, or other markers in the tumor and metastatic sites would have made this case report more valuable to the literature as it would provide potential for treatment or the start of finding out pathogenesis.

The authors reference cats, dogs and rabbits a lot, where as presumably other ungulates (even if not bovids) would be closer and more relevant, as there are several references of uterine adenocarcinoma in cervids and suids

Reference 2- check the year, it looks like it should be 2016: https://onlinelibrary.wiley.com/doi/book/10.1002/9781119181200

Author Response

The manuscript is well written.  It is true that there are very few published reports of uterine neoplasia in ungulates in general, and specifically in antelopes, therefore this case is valuable, but this is only one case.

We thank the reviewer for their appreciation of our “well written” manuscript and are delighted that they believe it represents a “valuable” case report. It is true this is “only one case”, however, due to the rarity of finding uterine neoplasia in ungulates in general, as the reviewer rightly pointed out, it was unfortunately not possible to present a case series.

The authors have done a great job of describing the findings and using CK IHC and TEM to confirm origin of metastatic lesions.

We thank the reviewer for their appreciation of our efforts in describing the details of the case and confirming the origin of the metastatic lesions.

However, it reads as a very long detailed histopathology report, and other than a confirmed metastatic uterine carcinoma in one sable antelope, there is little additional information in the manuscript.

We felt that the histopathology was a critical part of this case, as without this analysis, undoubtedly the diagnosis would have been a tumour of lymphoid origin based on the appearance of the spleen at autopsy. Thus, it was the histopathological diagnosis (with IHC and TEM) that confirmed the origin of the metastatic lesions and thus the primary tumour itself. This now adds another valuable case to the sparse literature of uterine neoplasia in ungulates, specifically in an antelope, and as such veterinarians can be more aware of this as a possible diagnosis when performing clinical examinations of antelopes.

The authors provide few details about the history of the female: is this a zoo and thus intensely managed, potentially contracepted? or a reserve where she presumably bred every year? Because the reproductive life history can influence risk factors please add some the missing history about this female

We contacted the owner to request this information, and it has now been added in. The sentence reads: “The sable antelope originated from a private game farm where the animals are kept in an extensive system consisting of large camps (20ha) with natural vegetation and receiving some supplementary feeding during the dry winter months. She was born on the farm and had given birth uneventfully to six calves on a yearly basis since reaching reproductive maturity in 2008 until her death in 2014”.

Upon close examination of her records the owner realised that she was born in 2005 and not 2006 as stated upon submission. Therefore, “eight-year-old” was changed to “nine-year-old” throughout the manuscript. Sorry for this mistake.

Line 256- in order for the reader to be able to appreciate that 8yr is an older female, please add the median life expectancy for this species

This information has now been added in. The sentence reads: “However, in both humans and animals uterine adenocarcinoma is generally seen in older females [2], as was the case with the sable antelope in this report (sable antelopes have a life-span of up to 16 years in the wild)”.

In the discussion the authors compare histological findings in tissues in the tissue as well as metastases.   Other IHC such as hormone receptors, or other markers in the tumor and metastatic sites would have made this case report more valuable to the literature as it would provide potential for treatment or the start of finding out pathogenesis.

No other IHC markers were included, since the histologic changes were suggestive of carcinoma and the pan-CK positivity and TEM results confirmed a diagnosis of epithelial origin. We appreciate that IHC for ER and PR would be nice, however, we request that additional IHC not be a requirement for publication as chose to follow characterisation of the tumour from a diagnostic point of view, rather than an investigation into potential disease aetiology (or treatment, since the sable was already dead).

The authors reference cats, dogs and rabbits a lot, where as presumably other ungulates (even if not bovids) would be closer and more relevant, as there are several references of uterine adenocarcinoma in cervids and suids.

We thank the reviewer for this excellent suggestion. Whilst we already have references to uterine adenocarcinoma in cattle in the text, we now have included eight references related to the additional relevant species requested, specifically deer and pigs.

Reference 2- check the year, it looks like it should be 2016: https://onlinelibrary.wiley.com/doi/book/10.1002/9781119181200

The hardcopy of the book says it was published in 2017, so we would like to keep it as that please. See screenshot below please.
